# Duration of Mentoring Relationship Predicts Child Well-Being: Evidence from a Danish Community-Based Mentoring Program

**DOI:** 10.3390/ijerph19052906

**Published:** 2022-03-02

**Authors:** Anna Piil Damm, Emma von Essen, Astrid Jæger Jensen, Freja Kerrn-Jespersen, Sarah van Mastrigt

**Affiliations:** 1TrygFonden’s Centre for Child Research, Department of Economics and Business Economics, Aarhus University, 8210 Aarhus, Denmark; apd@econ.au.dk (A.P.D.); asjaeger@econ.au.dk (A.J.J.); frejak@econ.au.dk (F.K.-J.); vanmastrigt@psy.au.dk (S.v.M.); 2Department of Sociology, Uppsala University, 751 26 Uppsala, Sweden; 3Department of Psychology and Behavioural Sciences, Aarhus University, 8000 Aarhus, Denmark

**Keywords:** child well-being, youth mentoring, duration

## Abstract

While a substantial body of literature suggests that lasting community mentoring relationships can have a range of positive effects on youths, little is known about these effects in the Nordic welfare context, where community mentees may have lower risk profiles compared to many previous samples. This study explores how the duration (length) of child mentoring relationships predicts parental perceptions of child well-being among 197 children served by Denmark’s most extensive community-based youth mentoring program. We find that children who have had a mentor for at least one year are perceived to have significantly higher well-being. In contrast, we find no significant differences in well-being between children who had mentors for less than one year and children on a waiting list. Previous research, conducted in primarily North American contexts, finds that longer mentoring relationships substantially improve school behavior and reduce risk taking. Our results add to the literature by indicating that a minimum mentoring relationship duration of one year appears to be similarly important in promoting well-being for youths involved in community-based mentoring programs in a Nordic welfare context.

## 1. Introduction

Mentoring initiatives have been shown to have multifaceted and broad impacts on youths, including improved social and academic achievement, reduced problem behavior, and improved psychological and physical well-being [1,2,3,4,5,6,7,8,9,10].

Youth mentoring programs have traditionally sought to improve the lives of children and youth by providing a relationship with an older, more experienced adult who is a good role model. Building a strong mentor–mentee relationship is likely to take time [11], and there is empirical evidence that a longer lasting relationship is positively associated with both relationship quality and perceived program support [12]. Moreover, early termination of a friendship may have negative effects, exacerbating negative views the child or youth may hold of themselves [13]. Therefore, many youth mentoring programs aim at establishing long-term relationships. In investigations of the effects of a youth mentor, it is therefore key to distinguish between effects achieved in the context of shorter versus longer relationships. Early qualitative studies suggested that the mentoring relationship needs to last for at least six months to have an effect [14]. Across various forms of youth mentoring, quantitative studies have further confirmed that longer, compared to shorter, relationships seem to be associated with a higher positive impact on youth academic, social and psychological outcomes [1,3,4,13,15,16,17].

For example, the seminal study by Grossman and Rhodes [13] found that youth in mentoring relationships that lasted a year or longer reported improved academic, psychological and behavioral outcomes compared to youth on a waiting list (those that had not yet been mentored). These effects became smaller when comparing youths on the waiting list to youths that had been mentored for a less than 12 months. Closest to our setting is a community-based mentoring study conducted in Canada [15] that found a decrease in emotional and behavioral problems for matches that lasted 12 months compared to non-mentored young people.

Most research on youth mentoring focuses on programs in the U.S. [18], most of which serve relatively high-risk communities. To date, little is known about potential positive or negative consequences of community-based youth mentoring in the context of the Nordic welfare state, which may be unique. Under the Danish Social Services Act Paragraph 52, children deemed to be at risk and in need of support must receive a social intervention implemented by the local government. This can include placement in out of home care or alternative preventive interventions, such as the provision of a mentor employed by the local government. Between 2016 and 2020, around 3.2% of the Danish population aged 0 to 22 received such a social intervention [19]. During 2016 and 2020, around 0.9% was removed from the home annually, while an increasing share from 2.6% in 2016 to 2.8% in 2020 received a preventive social intervention. Because the Danish government has a responsibility to implement social interventions for highly vulnerable children by, among other things, offering professional mentoring services, community-based volunteer youth mentoring programs do not typically serve the highest risk children in society. In the absence of social welfare provisions, community mentoring organizations in other national and cultural contexts are likely to serve a different, and possibly higher-risk, population. For example, Herrera et al. [20] note that the American Big Brothers Big Sisters program tends to reach “higher risk” children, where a large proportion has behavioral problems. The extent to which findings regarding the impacts of youth mentoring duration in those contexts generalize to the Nordic setting, is therefore largely unknown. To our knowledge, there are only two studies investigating effects of match duration in a Nordic welfare context, both of which evaluate the effects of programs targeting high-risk groups; one evaluates a school-based addiction prevention program [21] and the other is a qualitative evaluation of a community-based program for women with mental illness [22]. These studies suggest that match duration is positively associated with relationship quality [21] and that long lasting relations can improve the mental health of young females. However, better understanding of how volunteer youth mentoring impacts more general community samples in the Nordic countries is still needed.

### 1.1. The Current Study

In contrast to existing studies of the effects of having a community-based youth mentor, our study investigates impacts on Danish children likely to have a lower overall risk profile than many other mentee samples. Specifically, we investigate the impacts of having a youth mentor (referred to as an ‘adult friend’) provided by the largest community-based youth mentoring program in Denmark, Children’s Adult Friends (CAF). As part of a larger research project on the effects of having an adult friend, Perregaard [23] has previously conducted a qualitative fieldwork study following friendships made in this program from the time of the match and up to one year later. An important finding from that work is that the first year of the lasting friendships develop through four phases from ‘honeymoon’ to ‘a steady friendship’. This study supplements that work with a quantitative analysis of duration effects in a larger sample of children served by the same program.

In view of the qualitative findings by Perregaard [23] and aforementioned research on youth mentoring, we assume that adult friends are likely to improve child well-being by being good role models, by helping children to cope with school and peers, and by encouraging them to engage in socially acceptable and fun activities that allow them to reach their full potential. In line with previous research, we thus expect that longer length of a friendship with an adult friend will positively impact the child’s well-being, as the adult friend’s commitment, activities, and care are likely to increase the child’s quality of life.

However, in the Danish context, it is also conceivable that shorter friendships could have a positive impact on potentially lower risk children. Being matched with an adult friend could (in itself) have a positive impact on the child’s well-being because the knowledge that a resourceful adult wishes to be friends with the child is likely to boost the child’s self-esteem, while early termination of the relationship may have detrimental effects on child well-being as it may lower the child’s self-esteem. In a previous report and study, we found reciprocity to be a primary motive for volunteering to be an adult friend in CAF [24,25]. The immediate focus on the reciprocal aspect of the relationship by the adult friend also suggests that there might be a positive impact from shorter friendships.

Our hope is that results of this study can help to clarify the generalizability of previous duration findings and assist local associations in determining the best framework for establishing friendships that benefit children, and that are tailored to the specific profile of program participants and minimum beneficial length of a friendship.

### 1.2. Organizational Context—Children’s Adult Friends (CAF)

CAF was established in 1990 under the Danish name “Børns Voksenvenner” by a former social worker, who was inspired by the American mentoring program Big Brothers Big Sisters (BBBS). In contrast to BBBS, which started as a reaction to an increase in juvenile crime (www.bbbs.org/history, accessed on 12 January 2022), CAF was established as a response to the increase in divorces in Denmark, which often separated children from one of their parents [26]. The organization’s overall goal is to create a framework for friendships between children with a sparse family network and resourceful volunteer adults, typically of the same sex. Children can be offered an adult friend if the organization assesses that the child’s social bonds cannot be immediately strengthened through other relations.

CAF is the oldest and most centralized voluntary mentor program in Denmark. Today, it is part of two international networks of NGOs offering managed mentoring programs for children: Big Brothers Big Sisters International and the European Network of Children and Youth Mentoring Organizations. Since CAF operates in the context of the Nordic welfare model, however, the profile of the children and adult friends involved in the program may differ from BBBS, which targets higher risk children. As noted earlier, children in CAF are unlikely to have highly extensive social support needs because such children should instead receive support from their local government. If parents of children with psychiatric diagnoses and major behavioral challenges contact CAF, they are typically screened out.

CAF has a national office in the capital of Denmark, Copenhagen, and 42 smaller offices across Denmark, organized at the time of data collection around six regional offices (located in the larger cities) and several additional local offices (located in smaller towns). All regional and local offices follow extensive screening and matching processes of mentee families and mentors. CAF’s main task is to screen children, parents, and adult friends to make a qualified match.

## 2. Materials and Methods

Survey data for this study were collected in May–June 2017 from parents with children matched to an adult friend and parents to children on a waiting list for an adult friend.

The study population was constructed using information on registered users of the CAF program. We asked each of the local associations to send us contact information for families served by their local office. Prior to sending contact details to the research team, each local association sent an email to potential respondents allowing them to opt out of sharing their information for research purposes. All local associations that provided contact details were entered into a draw for 10 prizes of DKK 2500 (USD 380), recognizing the time commitment involved in providing this information. Contact lists were received from 33 out of 42 local associations, including all regional offices.

The resulting study population comprised 604 parents, with 21% waiting to be matched. Using the contact details provided by CAF, one parent per child was invited by email to participate in the study and provided with a direct link to an electronic survey. The parent who answered the survey is assumed to be a primary caregiver of the child, and to hold the most valuable knowledge of the child’s well-being. Even though parental assessments may differ slightly from children’s self-assessments, since it is the parent that has signed up the child for the program, we consider the parental assessment to be most relevant in an evaluation. Parents with several children on the waiting list or in a CAF friendship were asked to answer the questionnaire for one of their children (randomly selected by the research team). In both the introductory email and at the beginning of the questionnaire, respondents were informed that their participation in the survey was voluntary, and that individual responses would not be shared with CAF. Written informed consent was obtained from all subjects involved in the study. Prior to data collection, the study was registered with the regional Danish ethics board. In line with Danish law, as the board did not consider the study design invasive or sensitive, a full ethical review was not required.

To achieve as high a response rate as possible, all parents who completed the questionnaire were entered into a draw for 60 gift cards of DKK 300, 500 or 1000. We also sent email reminders to non-responders at regular intervals. In total, questionnaire responses were received from 230 respondents (38%), of which 197 had complete data for the main variables analyzed in this paper (33%). Due to data limitations, we cannot conduct an in-depth investigation of potential systematic non-response. Our data only allow us to compare demographic characteristics between children of respondent parents and the total population of CAF children. Compared to children in the full CAF population, the children of respondent parents in our final sample of 197 were slightly younger (43% vs. 58% over the age of 10), and were more likely to be male (72% vs. 68%).

### 2.1. Measures

The survey aimed to measure the respondent’s demographic and socioeconomic background, characteristics of the CAF friendship, and parental rating of their children’s well-being.

The child and parental demographic characteristics used are child’s gender, age at survey, and birth country as well as parental economic characteristics, civil status, number of children living at home, birth country and age at survey. We control for these variables as potential confounds.

To measure duration, we use the number of months the friendship had lasted at the time of data collection (mean = 26 months; median = 15 months). For some of the analyses, duration was further categorized into four intervals for: (a) children on the waiting list (16%), (b) children with a friendship of 1 to 6 months (18%), (c) children with a friendship lasting 7 to just under 12 months (10%), and d) children with a friendship of 12 months or more (56%). We employ three open source and internationally validated instruments used in previous mentoring research to measure well-being. While universal consensus regarding the specific definition of child well-being is lacking, the term can generally be understood to refer to collective “determinants of a good life for children, the promoters of growth and development, and factors that enhance a child’s feelings of happiness and satisfaction with life” [27]. This broad and multidimensional conceptualization encompasses a range of indicators including physical health and safety, economic status, education, mental and emotional health, and social relationships, e.g., see UNICEF [28]. In this paper, we focus on psychological, scholastic and social elements of well-being that might be affected by an adult friend. Specifically, we aim to capture both positive and negative manifestations of well-being reflecting both strengths and challenges experienced by the children and youth in our sample. To address critiques leveled against unidimensional approaches [29] we use multiple well-being measures:

*Strengths and Difficulty Questionnaire* (SDQ): A validated Danish version of the SDQ (parental report, designed for children aged 11–17) was used to measure general well-being among the children in our sample. The SDQ has good psychometric properties and is widely used internationally to explore children’s emotional problems, behavioral difficulties, hyperactivity/inattentiveness, problems in relation to peers, and social strengths [30,31]. The standard SDQ measure contains 25 questions rated on a 3-point scale from Not true ‘0′, Somewhat true ‘1′ to ‘ Certainly true ‘2′, rated according to the extent to which the statement fits the child (e.g., I get very angry and often lose my temper). These items are supplemented by a further seven questions concerning the impact of the child’s difficulties (e.g., Do the difficulties upset or distress your child?). We follow Goodman, Lamping and Ploubidis [32] and compute four subscale scores: (i) an ‘internalizing subscale’ (a combination of scores for emotional and behavioral difficulties, Cronbach’s alpha: 0.78), (ii) an ‘ externalizing subscale ‘ (a combination of scores for conduct and ‘hyperactivity/attention difficulties’, Cronbach’s alpha: 0.81), (iii) a social strength subscale (comprising the five strength questions, Cronbach’s alpha: 0.56), and (iv) a subscale measuring the impact of the difficulties (Cronbach’s alpha: 0.71). For each subscale, higher scores indicate greater difficulties or strengths [33]. The possible ranges of the subscale scores were 0–20 for externalizing and internalizing problems, 0–10 for the impact scores, and 0–10 for social strengths.

*Child and Youth Resilience Measure* (CYRM-PMK designed for ages 9–23, [34]) was used to measure parental ratings of their children’s resilience, i.e., the individual’s own ability to achieve positive results despite adversity. Resilience is not perceived as a static personal characteristic, but rather as a social and dynamic construction [35]. The CYRM includes items measuring individual resources (personal skills, peer support and social skills), relation to the primary caregiver (psychological caregiving), and contextual resources (education and cultural). Due to Cronbach alpha scores under 0.5 for the caregiver and contextual subscales, we only include the individual resources subscale in our analyses (11 items, Cronbach’s alpha: 0.78). Each question had three possible response categories, which indicated the presence of robustness-promoting resources (‘No = 1′, ‘Sometimes = 2′, ‘Yes = 3′). The possible range of the subscale scores for individual resources was 11–33, where a high value is an expression of a more robust child.

*Self-Perception Profile for Children* (SPPC, adult rater version, designed for children aged 8–14 [36,37]) is a multidimensional scale that measures child competencies across five specific areas: perceived academic competencies, social skills, athletic skills, physical appearance, and behavior. In this study, we use parental ratings judging academic (Cronbach’s alpha: 0.73), social (Cronbach’s alpha: 0.80), and athletic competencies (Cronbach’s alpha: 0.68). For each subscale, we asked parents to indicate the extent to which double statements such as the following fit their children: This child finds it hard to make friends OR For this child it’s pretty easy to make friends. The answer categories were ‘Really true’ or ‘Sort of true’ for each part of the statement, where the first part of the statement was coded as ‘Really true = 1′ and ‘Sort of true = 4′, while it was coded as 2 and 3 for the second part for the statement. For each area of competency, we calculated an average subscale score ranging from 1 to 4, where higher scores reflect parents’ assessment of their child’s competencies in relation to each area.

### 2.2. Analytic Approach

The survey data were first examined and condensed to a more usable form: open answers were encoded in appropriate categories based on Statistics Denmark’s categories of, for example, educational attainment (variable HFUDD11), occupation (variable BESKST02) and country of birth (categories with few observations were combined with other appropriate categories or removed based on anonymization requirements). To gain efficiency in our regression analysis, rather than dropping observations with missing values for birth country and socioeconomic background characteristics, we coded such missing values as zero and included indicator variables for the missing value of each of these control variables. However, 33 cases with missing data regarding duration of friendship or any of the well-being outcomes were necessarily excluded from analysis, rendering a final analytic sample of 197 respondents with complete data on match duration, SDQ, CYRM and SPPC.

Our main hypothesis is that child well-being will increase with longer match duration because it takes time to build strong friendships. A second hypothesis is that being matched per se improves child well-being, as the knowledge that a resourceful adult wishes to be friends with the child is likely to boost the child’s self-esteem.

Empirically, we test these hypotheses using multivariate regression analysis. For each dimension of child well-being (measured using the aforementioned subscales of SDQ, CYRM and SPPC), we regress the measure of child well-being on match duration using two different specifications of match duration in months: (i) a continuous variable for match duration, allowing for heterogeneous effects across gender, and (ii) a categorical measure based on four duration intervals: 0 (waiting list), 1–6 months, 7–11 months, and ≥12 months, where the waiting list group is used as the reference category. While tests based on the first specification allow us to draw conclusions about the overall effect of friendship duration, the results of our second specification allow us to identify potential duration thresholds for positive effects. For both specifications, a positive and statistically significant association between match duration and the child’s well-being can be interpreted as consistent with our hypotheses.

Given that demographic and socioeconomic background characteristics are likely to correlate statistically with child well-being as well as with match duration, we report all results controlling for child gender and age (at the time of survey completion), as well as parental socioeconomic background and demographic characteristics.

Since the empirical analysis uses an observational data approach rather than an experimental design, the coefficient estimates of match duration on measures of child well-being should be interpreted as statistical associations rather than be given a causal interpretation. However, the statistical associations can provide suggestive evidence on these effects.

## 3. Results

Summary statistics of individual demographic and socioeconomic characteristics for the full regression sample are shown in the first column of Table 1. With few exceptions, the survey was completed by the mother. The responding parent was typically single and between 36–50 years at the time of the survey (mean age of 43). The overwhelming majority had a vocational or tertiary education and around two-thirds were employed, while a smaller proportion were unemployed or students. Nearly three quarters of the children were boys. The children were 10.6 years old on average, and typically lived with their (single) mother, and usually 1–2 siblings. Approximately 1 in 10 of the responding parents had immigrated to Denmark from abroad.

Table 1 also reports summary statistics for three mutually exclusive subgroups of the regression sample: children who had an adult friend for at least 12 months at the time of the survey, children who had been matched with an adult friend for less than 12 months at the time of the survey, and children on the waiting list. We regard the former as the treatment group since the CAF program aims at providing children with a relationship of at least one year. Children on the waiting list for an adult friend constitute the reference group in the regression analysis. As illustrated in Table 1, children on the waiting list for an adult friend have a similar socioeconomic background to the group of children who have had an adult friend for less than a year, although children on the waiting list for an adult friend includes a higher proportion of boys. This demographic difference reflects the practice that children are generally matched with an adult friend of the same sex as they reach the top of the waiting list, and the fact that boys on average wait longer for a match because of a lack of male volunteers. The similarity in socioeconomic background for children on the waiting list and children who have been matched for less than a year supports our decision to use children on the waiting list in the regression sample as our reference group.

Well-being profiles of the children in our regression sample and duration subgroups are shown in Table 2. The means and standard deviations observed for our regression sample point to a group of children with slightly elevated risk compared to available general population means. For the SDQ [33] there are population norms estimated for both the U.S. and Denmark, using representative samples of children in different age spans. Unfortunately, the age spans for the U.S. and Danish norms differ, invalidating a direct comparison. The age span for the Danish norms is also not the same as the age span in the current regression sample. However, in the technical report from the full survey on which this study is based [24], we constructed a limited sample of the same age span as for the Danish norms (children aged 5–7 and 10–12) comparing the SDQ scores. We concluded that the children in our survey had, on average, higher internalizing and externalizing difficulties and impacts from these difficulties compared to Danish children in general. Focusing on social strengths, however, we find that the children from the full survey on which this study is based, have similar social strengths to Danish children in general [33].

The age span used for the U.S. norm sample was unfortunately of 11–14, but these norms display higher SDQ difficulties compared to the Danish norms (children aged 10–12). De Wit et al. [38] find children aged 7–14 years in BBBS to have SDQ scores well above the means of difficulties scores among the U.S. representative samples (children aged 8–10 and 11–14).

In sum, based on the SDQ scores for the regression sample in Table 2, the children in CAF seem to be of higher risk than the general Danish population in terms of difficulties, but our CAF sample appears to have similar social strengths to children in general in Denmark.

If we turn to the children’s resilience scores, our observed means are high considering the ranges of the scales, suggesting that CAF treats a rather robust sample of children. The mean score on the individual subscale of the CYRM in the regression sample is approximately one standard deviation below the maximum score. The SPPC scores are all above the midpoint of the scale, in line with the original sample of U.S. children by Harter [36]. The children in the regression sample have the highest perceived scholastic competencies, slightly lower social abilities and lowest perceived athletic competencies.

Turing to the mean well-being scores across subsamples of children on the waiting list, with a match less than 12 months, or with a match of at least one year, a consistent pattern emerges, such that children with longer match durations generally appear to demonstrate fewer difficulties and greater strengths and competencies. In the next section, we test whether these differences are significant when controlling for child and family background.

### Regression Results

We first present ordinary least squares (OLS) regression results exploring the influence of friendship duration on children’s strengths and difficulties as measured by SDQ and assessed by the respondent parent. In Table 3, we report the coefficient estimates from OLS regressions for each of the four SDQ subscales: (i) internalizing problems, (ii) externalizing problems, (iii) social strengths, and (iv) impact of difficulties. Recall that a higher scores on SDQ subscales indicate more problems, except for the subscale for social strengths for which a higher score indicates more strengths. Panel A reports the coefficient estimates using our first specification, in which friendship duration is measured as a continuous variable from 0 months to the maximum friendship duration in months among individuals in the regression sample. This specification allows for heterogeneous effects of friendship duration across the child’s gender by interacting the duration variable with separate indicators for male children and female children. These coefficients come from joint estimation and indicate the effects of duration for each gender. F-tests of equal coefficients formally test whether the statistical association between friendship duration and the outcome is the same for boys and girls (see Table notes). The coefficient estimates of additional control variables are not reported but available upon request.

The regression results in Table 3, Panel A show a negative and statistically significant association between each additional month of adult friendship duration and the internalizing score for both boys and girls. According to the point estimates, each additional month of an adult friendship is associated with an internalizing score 0.03 points lower for boys and 0.09 points lower for girls. The coefficient estimate is three times larger for girls than boys, indicating a greater decrease in girls’ internalizing problems for each additional month of adult friendship; the coefficient estimates are significantly different from each other. Assuming linear effects of friendship duration, boys who have had 12 months of adult friendship score 0.4 lower on the SDQ scale of internalizing problems, while the corresponding effect is 1.1 for girls. The coefficient estimates of an additional month of adult friendship duration on the SDQ externalizing scale, social strengths and impact are all statistically insignificant, irrespective of gender.

Turning to the regression results reported in Table 3, Panel B, showing coefficient estimates based on our categorical specification of duration, our findings indicate the existence of a threshold of at least 12 months of friendship duration for significant impacts. Children who have had an adult friend for at least 12 months on average score 1.3 lower on the internalizing problems subscale compared with children on the waiting list, although the difference is not significant at a conventional 5% significance level. The estimate of 1.3 corresponds to 22% around the mean score in the regression sample. Moreover, children who have had an adult friend for at least 12 months on average score 0.9 lower on the impact subscale; the estimate is significant at a 5% level compared with children on the waiting list, corresponding to 65% lower impact score at the mean score. On average, the four duration groups of children have similar levels of externalizing problems and social strengths.

Taken together, the results in Table 3 provide suggestive evidence that longer durations of adult friendship reduce internalizing problems, especially for girls, and that friendships of at least one year are important for reducing the impacts of such difficulties.

Table 4 presents the results from OLS regressions investigating the role of friendship duration on child robustness promoting resources as measured by CYRM, using the same specifications of friendship duration as in Table 3. Recall that a higher CYRM-subscale score indicates greater robustness promoting resources. The coefficient estimates shown in Panel A suggest that children with longer adult friendships have improved individual capacities/resources. Here, one additional month of friendship is associated with a capacity score 0.04 higher for boys and 0.05 higher for girls. These coefficient estimates are not significantly different across gender. If we assume a linear effect of duration, 12 months of additional adult friendship is associated with a score on individual resources which is 0.5–0.6 points higher, corresponding to 2% around the sample mean.

The results in Panel B again indicate a threshold effect, such that children who have had an adult friend for at least 12 months have significantly better individual resources compared with children on the waiting list. The coefficients are positive, showing an increase of 1.6 points, corresponding to an increase of 5.7% around the mean of the sample.

Overall, the results in Table 4 suggest that adult friends promote the children’s individual resources across gender, particularly for friendships that last at least 12 months.

Table 5 presents the results from OLS regressions exploring the impact of duration across three subscales of the SPPC: (i) athletic competencies, (ii) scholastic competencies and (iii) social competencies. As in Table 3 and Table 4, Panel A reports coefficient estimates of duration in months for boys and girls, whereas Panel B reports coefficient estimates for each time interval (1–6 months, 7–11 months and at least 12 months) compared to children on the waiting list. Assuming a linear effect of friendship duration on parental rating of child competencies, and allowing for heterogeneous effects across child gender, the coefficient estimates reported in Panel A show that children with longer adult friendships score significantly higher on the SPPC subscales measuring athletic and social competencies than children on the waiting list. According to these estimates, children with a friendship of at least 12 months score 2–4% higher on athletic competencies and 3–5% higher on social competencies. The estimates are not significantly different for boys and girls and the statistical association between friendship duration and scholastic competencies is insignificant.

The results in Panel B show that children who have had an adult friend for at least 12 months score significantly higher on athletic and social competencies. The coefficient estimates are around 0.33, corresponding to 13% higher athletic competencies and 11% higher social competencies, although the coefficient estimate on social competencies is not significant at a conventional 5% significance level.

Overall, the positive associations between match duration and child well-being presented in Panels A of Table 3, Table 4 and Table 5 lend support to our first hypothesis, that child well-being would increase with match duration. Additionally, the threshold effects shown in Panel B of the same tables indicate that a match duration of at least 12 months is crucial to promoting child well-being, with little effect of shorter friendships. This suggests that simply being matched is not enough to improve a child’s well-being. We thereby fail to support our second hypothesis.

## 4. Discussion

The results of our regression analysis are in line with previous literature on community-based youth mentoring programs in the North American context, e.g., [10,12,13,15], and support our first hypothesis that longer lasting friendships are beneficial to child well-being. We contribute to making the understanding of youth mentoring more diverse by presenting results from the Nordic welfare state context. Here, where the local governments are obliged to offer social interventions targeting children of higher risk, the children in the Danish community-based youth mentoring program CAF had higher difficulty scores than similarly-aged peers in the full population, but similar social strengths scores as measured by SDQ. We interpret this as evidence that the children in the CAF program tend to exclude those in the highest risk group. Among these lower risk children, we find significantly higher well-being among those who have had their mentor for at least one year compared to those on the waiting list.

More specifically, our results show improvements in the child’s internalizing problems, social and athletic competencies, and individual resilience/robustness. In addition, we find that the impact of an adult friend on well-being only appears after one year of adult friendship (comparing with children on the waiting list). By contrast, we find no evidence that an adult-friendship of at least one year improves children’s externalizing behavior or scholastic competencies.

Our second hypothesis that a match in itself (regardless of duration) might improve child well-being, e.g., by boosting self-esteem, was not supported by our data. This could be because it takes time to build a trusting friendship. We believe an adult friendship positively influences the child’s well-being because the adult friend’s commitment, activities, and care are likely to increase the child’s quality of life. The duration of the friendship is associated with more well-being as the quality of the friendship is continuously being strengthened [39]. One possible explanation for this duration effect is that the adult friend better gets to know the child’s strengths and difficulties over time and can thereby organize activities that may both develop the child’s strengths and address the child’s challenges (while having a good time). Another possible explanation is that as the child gets to know the adult friend better, the child might agree more to trying the activities that the adult friend suggests. The improved social competencies and individual resilience/robustness may be due to greater adult contact and attention, which affects the child’s self-esteem and perceived social acceptance. The regression results also show that friendships lasting one year or longer positively influence the child’s athletic skills. This could, again, be related to the child participating in more active and exploratory activities with the adult friend as the relation evolves, including activities that may not be typical of the child’s own family due to fewer social and economic resources.

As noted earlier, some studies suggest that adolescents who were in relationships that terminated within a very short period reported declines in several indicators of general functioning [13]. Accordingly, the goal of the CAF organization is to establish friendships that last at least one year. In our context, friendships that terminate before one year as opposed to those that have not yet reached one year are considered to have ended prematurely. A limitation of this study is that it was not possible to directly test the effects premature terminations, given that only 5% of friendships in our sample ended prematurely. Credible examination of the effects of an adult friendship with premature termination would require exogenous variation in adult friendships. ¨ The fact that we find a minimum one-year threshold for positive impact (which corresponds with the minimum desired length of CAF friendships) raises an interesting question of whether duration thresholds may be context-dependent. That is, whether child outcomes might be impacted by whether expectations regarding the minimum desired duration of friendship are met or not. Future research exploring this question would be valuable to further advance understanding of duration effects like those observed here.

Another limitation of our study is the low response rate and data limitations that prevent us from conducting an in-depth investigation of non-response. However, comparing demographic characteristics between children of respondent parents and the full population of CAF children, we see only modest differences.

Our study is further limited to the evaluation of the influence of having an adult friend on parental assessment of child well-being. It is the parents who apply to CAF for an adult friend for their child. Given that our analyses investigate the effects of an adult friend on child well-being as assessed by the parent, it likely corresponds to the relevant person in the household evaluating the child’s well-being. Parents may, on the one hand, be (optimistically) inclined to perceive benefits of the program they have chosen for their child, thus overstating child well-being after match. On the other hand, parents may understate child well-being to maintain program support. Similar analyses based on child reports would therefore be useful.

Lastly, we use children on the waiting list as our main comparison group. Some of these children might never be matched with an adult friend. The comparison group may thus differ in observable and unobservable characteristics compared to the children who are or become matched. Since we find children on the waiting list to have similar socioeconomic backgrounds as children who have been matched for less than a year, we considered the children on the waiting list to be an appropriate reference group in the regression sample. An important avenue for future research is to explore more elaborate control group designs to understand the effects of an adult friend on various outcomes of children with elevated difficulties, but who have average social strengths, as in this sample.

Notwithstanding these limitations, we present the first suggestive evidence in the Nordic welfare context of the effect on child well-being of having an adult friend for at least one year. These results contribute to a more nuanced understanding of international duration effects and may help local associations to determine the most favorable frameworks for an adult friendship. In line with previous work in higher risk samples, these results confirm that organizations should require a minimum commitment of at least one year from volunteer mentors in order to ensure that the adult friendship has the desired positive significance for the child’s well-being.

## 5. Conclusions

In a community-based youth mentoring program in a Nordic setting, we explore a sample of children that have higher internalized and externalized problems, but the same level of social strengths, compared to their similarly aged peers in the general population. Our regression results suggest that although having a volunteer adult friend does not appear to impact externalizing behavior or scholastic competencies, children with a longer duration of mentoring relationships display fewer internalizing problems, greater social and athletic competencies, and higher individual resilience/robustness. Further, our findings suggest that steady friendships of at least one year are crucial to producing these positive effects. These findings confirm that long-lasting community-based youth mentoring relationships are also beneficial for children’s well-being in the context of the Nordic welfare state. Carefully screened volunteer mentors who are willing to commit to being an adult friend for at least one year may therefore have an important role to play in providing a wide range of children and youths with a stronger social network. An important consideration for future research and policy in this area is to explore how best to limit the number of children with weak family and social bonds who need an adult friend. That is, to further consider both causes and solutions to the lack of adult support in some families.

## Figures and Tables

**Table 1 ijerph-19-02906-t001:** Summary Statistics of Individual Demographic and Socioeconomic Background Characteristics.

	Regression Sample		Friendship Duration (≥12 Months)		Friendship Duration (<12 Months)		Waiting List	
	N	Mean (SD)	N	Mean (SD)	N	Mean (SD)	N	Mean (SD)
Child characteristics:								
Age	197	10.60 (3.33)	119	12.14 (3.42)	68	9.12 (2.39)	35	9.17 (2.87)
Male	193	0.72	116	0.75	67	0.63	35	0.77
Parent characteristics: ^a^								
Age	181	42.77 (6.57)	95	44.40 (6.48)	58	41.02 (6.06)	31	41.19 (6.66)
Female	187	0.98	99	0.97	59	0.98	32	1.00
Highest educational level								
Upper secondary education	187	0.31	99	0.31	59	0.32	32	0.25
Tertiary education	187	0.58	99	0.60	59	0.58	32	0.56
Employed	187	0.63	99	0.70	59	0.59	32	0.44
Single	187	0.72	99	0.72	59	0.73	32	0.69
More than one child living at home	187	0.94	99	0.95	59	0.95	32	0.91
Danish origin	187	0.91	99	0.94	59	0.88	32	0.91

Note: ^a^ Refers to the parent who answered the survey.

**Table 2 ijerph-19-02906-t002:** Means and Standard Deviation of the Eight Well-being Outcomes Across Samples.

	Regression Sample	Friendship Duration (≥12 Months)	Friendship Duration (<12 Months)	Waiting List
	Mean (SD)	Mean (SD)	Mean (SD	Mean (SD)
SDQ Internalizing Problems	6.00 (4.20)	5.34 (4.03)	6.83 (4.47)	6.18 (3.96)
SDQ Externalizing Problems	5.66 (3.93)	5.11 (3.85)	6.52 (4.19)	5.56 (3.29)
SDQ Social Strengths	8.46 (1.53)	8.38 (1.59)	8.60 (1.56)	8.50 (1.29)
SDQ Impact of Child Difficulties	1.37 (2.06)	1.11 (1.80)	1.61 (2.29)	1.59 (2.24)
CYRM Individual Capacities/Resources	27.97 (3.53)	28.70 (2.96)	27.34 (3.97)	27.26 (3.75)
SPPC Athletic Competencies	2.61 (0.75)	2.71 (0.81)	2.50 (0.67)	2.50 (0.75)
SPPC Scholastic Competencies	3.21 (0.78)	3.23 (0.79)	3.22 (0.72)	3.23 (0.82)
SPPC Social Competencies	2.89 (0.81)	2.97 (0.80)	2.82 (0.80)	2.80 (0.89)
N	197	103	63	34

**Table 3 ijerph-19-02906-t003:** Results From Ordinary Least-Squares Regression of SDQ Subscales on Duration of Friendship.

	Internalizing Problems		Externalizing Problems		Social Strengths		Impact	
	Estimate	SE	Estimate	SE	Estimate	SE	Estimate	SE
Panel A: Duration in months allowing for heterogeneous effects across child gender								
Intercept	8.275 ***	2.713	13.510 ***	2.530	8.192 ***	1.032	0.975	1.331
Duration (male)	−0.030 **	0.014	−0.001	0.013	−0.004	0.006	−0.008	0.007
Duration (female)	−0.090 ***	0.029	−0.029	0.027	−0.006	0.011	−0.021	0.014
R^2^	0.282		0.284		0.219		0.281	
Panel B: Duration in 6 months intervals								
Intercept	9.197 ***	2.867	13.680 ***	2.590	8.052 ***	1.054	1.305	1.361
Duration: 1–6 months	−0.118	1.075	1.171	0.971	0.332	0.395	−0.363	0.512
Duration: 7–11 months	−0.298	1.266	−0.345	1.143	0.200	0.465	−0.436	0.601
Duration: At least 12 months	−1.267	0.944	0.405	0.853	−0.192	0.347	−0.891 **	0.449
R^2^	0.240		0.289		0.228		0.287	

Note. N = 197, except for regression on social strengths (N = 195). *** *p* < 0.01, ** *p* < 0.05, **^+^** *p* < 0.1. Standard errors clustered by the local office (8 clusters: 7 regional offices and other local offices). The possible ranges of the subscale scores were 0–20 for the externalizing and internalizing problems, 0–10 for the impact scores, and 0–10 for the social strengths. The social strength score is the only SDQ score where a higher score is positive, as it reflects greater social strengths. All regressions control for child’s gender, age and birth country as well as socioeconomic characteristics of both parents and respondent parent’s civil status, number of children living at home, birth country and age. Regarding Panel A, the duration coefficients come from joint estimation, and the *F*-test of equal coefficients of duration across gender is rejected at a 5% significance level only for the outcome “Internalizing problems”. In Panel B, the reference group is waiting list.

**Table 4 ijerph-19-02906-t004:** Results of Ordinary Least-Squares Regression of CYRM Individual Capacities/Resources Subscale on Duration of Friendship.

	Individual Capacities/Resources	
	Estimate	SE
Panel A: Duration in months allowing for heterogeneous effects across child gender		
Intercept	26.820 ***	2.288
Duration (male)	0.037 ***	0.012
Duration (female)	0.045 ^+^	0.024
R^2^	0.274	
Panel B: Duration in 6 months intervals		
Intercept	25.440 ***	2.398
Duration: 1–6 months	0.788	0.899
Duration: 7–11 months	1.042	1.059
Duration: At least 12 months	1.618 **	0.789
R^2^	0.244	

Note. N = 197. *** *p* < 0.01, ** *p* < 0.05, **^+^** *p* < 0.1. Standard errors clustered by the local office (8 clusters: 7 regional offices and other local offices). The possible range of the subscale scores for individual resources was 11–33. A high value is an expression of a more robust child. All regressions control for child’s gender, age and birth country as well as socioeconomic characteristics of both parents and respondent parent’s civil status, number of children living at home, birth country and age. Regarding Panel A, the duration coefficients come from joint estimation, and the F-test of equal coefficients of duration across gender cannot be rejected at a 5% significance level. The reference group in Panel B is children on the waiting list.

**Table 5 ijerph-19-02906-t005:** Results of Ordinary Least-Squares Regression of SPPC Subscales on Duration of Friendship.

	Athletic Competencies		Scholastic Competencies		Social Competencies	
	Estimate	SE	Estimate	SE	Estimate	SE
Panel A: Duration in months allowing for heterogeneous effects across child gender						
Intercept	2.826 ***	0.502	2.849 ***	0.534	2.847 ***	0.534
Duration (male)	0.008 ***	0.003	0.002	0.0028	0.006 **	0.003
Duration (female)	0.003	0.005	0.0003	0.006	0.013 **	0.006
R^2^	0.231		0.200		0.259	
Panel B: Duration in 6 months intervals						
Intercept	2.631 ***	0.523	2.675 ***	0.548	2.502 ***	0.556
Duration: 1–6 months	0.124	0.196	−0.003	0.205	0.128	0.208
Duration: 7–11 months	0.117	0.231	0.203	0.242	0.363	0.245
Duration: At least 12 months	0.335 ^+^	0.172	−0.002	0.180	0.327 ^+^	0.183
R^2^	0.209		0.202		0.237	

Note. N = 197. *** *p* < 0.01, ** *p* < 0.05, **^+^** *p* < 0.1. Standard errors clustered by the local office (8 clusters: 7 regional offices and other local offices). Across subscales, the possible range of scores are between 1 and 4, where a higher value indicates better competencies in the domain. All regressions control for child’s gender, age and birth country as well as socioeconomic characteristics of both parents and respondent parent’s civil status, number of children living at home, birth country and age. Regarding Panel A, the duration coefficients come from joint estimation, and the F-test of equal coefficients of duration across gender cannot be rejected at a 5% significance level. The reference group in Panel B is children on the waiting list.

## Data Availability

All data from the survey are stored on a secure server owned by Aarhus University and will be made available for further research in anonymized form through the Danish National Archives (sa.dk) following publication of the full project. Moreover, we will publish our codes on the website of https://childresearch.au.dk/.

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
