# Peer review of "Duration of Mentoring Relationship Predicts Child Well-Being: Evidence from a Danish Community-Based Mentoring Program"

_ijerph, 2022, doi:10.3390/ijerph19052906_

Round 1
Reviewer 1 Report
Authors present a hard work in good english and clear tables.
Yet, the family contribution needs to be discussed further. Family bonds need to strenghen, do they? the solution of another adult is a sine qua non or hides the real problem? Please discuss.
How such an adult is chosen by the authorities? May this jeopardize democracy?
Reviewer 2 Report
1.Duration of mentoring relationship predicts child well-being is a very meaningful research topic, which can be used as a positive example for adults when facing adolescents with special needs and situations.
2.The issue of "Quality adult companionship is helpful for children's growth" has been accepted by the public. In this study, the main target is children who are listed in CAF institutions. However, in Materials and Methods, there is no specific age group. instruction. Internationally speaking, the age range of children is generally 0-18 years old. Therefore, whether the interaction between adults and children and the time spent with them are related to the age of children is also a very important issue, and it also affects the appropriate use of measures. It is suggested that the age group of the research subjects should be stated here, and the background age of the use of the three scales(SDQ/CYRM-PMK/SPPC) should also be stated.
3.(L194) here is only a brief description of the use of these three scales in previous studies and therefore also used in this study. However, from the selection of research tools, researchers should make a clearer definition of the "child well-being" to be discussed in this study, and then confirm that the three scales used can support the research purpose. It is suggested to add an explanation of the definition of "child well-being" and how these three scales support this study's understanding of child well-being
4.(L250)The number of children mentioned in the analytic approach was 604, and then the sample was reduced to 197. The article did not explain the basis for the selection of the final sample, please explain why these 197 were selected in this study
5.The conclusion of the study is too brief, and it is necessary to make a clearer conclusion explanation for the purpose of the study according to the obtained results to present the specific value of this study
Reviewer 3 Report
This manuscript, submitted to the International Journal of Environmental Research and Public Health, presents a study on the relationship between the duration of child-adult mentoring relationships and child well-being outcomes. This is an interesting topic that seems well supported from the literature and well presented in this manuscript. There are several repeating small language errors (incorrect positioning of indefinite articles, and frequent words with incorrect hyphens, see, for example, lines 61,82, 94, 166, etc.) Attention should also be paid to lines 206-211 where it states that there are three subscales and then presents four, and line 240, where “for the first part of the statement” is unnecessarily repeated from the line above. These, and other small errors, can be corrected with a careful revision and editing of the manuscript.
However, the presentation of the data and the results is confusing, so much so, that it is impossible to evaluate the veracity of the claimed findings. These sections need to be thoroughly rewritten, to present clearly and concisely the nature of the data and the findings, in both narrative and tabular formats (succinctly applying typical APA formats for tables of regressions analysis, etc.). In particular:
- Table A1 in the appendix is unnecessary – this information can be explained in text.
- Table 1 must be inserted close to its first mention in the text (or include the page number where it is inserted) and should provide summary statistics for only the most relevant variables. A table that is 1.5 pages long, and 16 columns wide, is not useful. I suggest removing the columns for full population and including these statistics in text, combining all mean and SD columns into 1 column, mean (SD), and collapsing the control variables into 1 statistic for each (i.e., average level of educational attainment), or just displaying the largest category for each (i.e., Employed %)
- Table 2 should detail the summary statistics for the outcome variables in the same format as Table 1 (with the same subgroups)
- As the outcome variables are formed from multiple items from multi-dimensional scales, there should be a more thorough discussion of these variables, including, typically, the Cronbach’s alpha for each. Because of the additional information that is required here, I would suggest reducing the manuscript to only focus on, present findings on, and discuss the most interesting of the outcomes – for example just the 3 or 4 outcomes that have significant findings.
- Then, with fewer outcomes to present, reduce tables 3-5 to tables that are more typically formatted for regression in APA, each table presenting 1 outcome, and the different analytic subgroups.
- Finally, discuss only what is relevant from these reduced tables.
Only with changes such as these, will it be possible to evaluate the quality of the analyses and findings of this study, and therefore make a recommendation for possible publication.
Round 2
Reviewer 3 Report
This is my second review of this manuscript, submitted to the International Journal of Environmental Research and Public Health, presenting a study on the relationship between the duration of child-adult mentoring relationships and child well-being outcomes. This is an interesting topic that seems well supported from the literature and well presented in this manuscript. The changes that the authors have made to the manuscript have hugely improved it, and it was now a pleasure to read and easy to understand. Congratulations on an excellent result from your hard work at editing.
I would like to mention just a few small changes that I would recommend before publication:
- In the abstract, try and be consistent with the use of either “youths” or “children” (whichever you think is most appropriate).
- On line 151 and 567 the term “sex” is used where I believe the term “gender” would be more appropriate.
- On page 5, where the Cronbach Alpha scores for each scale have been included (excellent!) it would be ideal to include the number of items as well for each scale. For example, “11 items, Cronbach’s Alpha .78” is a perfect example of how each should be reported.
- Line 411, “one in 10” should be changed to “1 in 10”
Thank you for the opportunity to review again this manuscript, and see the evidence of such hard and consistent editing work!